# Influence of the Level of Sex Hormones in the Blood of Gilts on Slaughter Characteristics and Meat Quality

**DOI:** 10.3390/ani13020267

**Published:** 2023-01-12

**Authors:** Krzysztof Tereszkiewicz, Łukasz Kulig, Piotr Antos, Karolina Kowalczyk

**Affiliations:** Departament of Computer Engineering in Management, Faculty of Management, Rzeszow University of Technology, al. Powstańców Warszawy 12, 35-959 Rzeszów, Poland

**Keywords:** pigs, gilts, sex hormone levels, progesterone, luteinizing hormone, meat quality

## Abstract

**Simple Summary:**

The authors aimed to investigate the possible impact of hormone levels during the reproductive cycle of gilts on the slaughter characteristics and meat quality. The levels of progesterone (P4) and LH were determined in the blood samples of the processed female pigs. Moreover, during the research, the biochemical and haematological properties of the blood were investigated. Additionally, the fat and meat content and other properties were determined. It was observed that some of the analysed properties of meat and blood varied as a result of increased levels of LH.

**Abstract:**

The aim of the conducted research was to determine the impact of factors such as sex hormone levels, which vary during gilts’ reproductive cycle, on the quality of the obtained meat and slaughter characteristics of the processed gilts. The research material included a population of 60 gilts slaughtered in one of the slaughterhouses located in south-eastern Poland. After the slaughtering operations were completed, the carcasses were weighed at the classification stand. The results of the statistical evaluation of the haematological and biochemical blood parameters of the examined gilts showed that, in the tested blood samples, the concentration of progesterone had a statistically significant impact only on the level of total protein, which was higher in the blood samples of gilts with a low concentration of progesterone. It was found that carcasses of gilts with higher levels of the LH hormone were characterized by a lower meat content index by nearly 3%. It was shown that the concentration of LH affected the post-slaughter temperature of the sirloin and ham muscles. The interpretation of the obtained data was difficult since there seems to be a gap in the literature concerning the dependencies of sex hormone levels in gilts and meat quality.

## 1. Introduction

Pork production is based on the fattening of gilts, boars, and male castrates. Boar castration can be performed by surgery or by immunization against gonadotropin-releasing factor (GnRF) [1]. The use of immunization with the Improvac vaccine significantly reduces the occurrence of boar taint, and additionally contributes to the improvement in production effects through better use of feed, protein deposition, and increase in the meatiness index of the carcasses [1]. In some countries, most producers now abandon the surgical castration of boars, considering it to be a procedure that causes pain and discomfort and significantly reduces animal welfare [2]. In extensive breeding systems, in order to avoid pregnancy and the related sanitary problems, gilts are sterilized [3].

Apart from the differences between the sexes, there is also a significant variation in slaughter parameters within the sex. As indicated by the results of the studies [4,5,6,7,8], the observed differences are mainly related to factors such as race, age at slaughter, and body weight.

According to the research presented in [9], the taste and smell of pork play a key role in consumer purchasing decisions.

These studies show that the carcasses of gilts, in comparison to the carcasses of male castrates, are characterized by lower fat content, a higher meatiness index, and greater meat-cutting efficiency. The authors [5,7] also indicate the influence of gender on the quality parameters of meat. The differences found mainly concern the colour of the meat, intramuscular fat content, and thermal leakage.

However, explaining the differences in the quality parameters of carcasses observed within individual sexes is still an unresolved issue. It can be assumed that the activity of the endocrine system related to reproductive functions may play some role in shaping the carcass features of gilts. At the age of 4–6 months, sexual maturity in gilts can be observed, which results in the appearance of the reproductive cycle, the length of which is on average 21 days with variations from 18 to 24 days and depends on the age and the maintenance system. According to some [6] studies, the sexual maturity of the gilts can be influenced by the intensive feeding system of fattening pigs. Gilts fed abundantly are characterized by higher levels of LH (luteinizing hormone), FSH (follicle-stimulating hormone), and oestradiol in the blood serum.

The reproductive cycle includes cyclically repeated phases: follicular, lasting 5–7 days, and luteal, lasting 13–15 days [10]. The cycle is regulated by the hormonal pathway of the pituitary gland and the neurohormonal pathway of the hypothalamus. The observed changes in the level of sex hormones are characteristic symptoms of phases of the reproductive cycle. During the follicular phase, low levels of progesterone are observed, the concentration of which increases in the luteal phase. On the other hand, in the pre-oestrus period, there is a significant increase in the level of oestradiol and the LH hormone [11,12]. Individual phases of the reproductive cycle are also accompanied by a number of characteristic behavioural symptoms, which mainly accompany the pre-heat and oestrus phases. In addition to the observed changes in behaviour during oestrus, gilts have an increase in motor activity and loss of appetite [13], increased nervous excitability, and an increase in body temperature. As can be anticipated, behavioural changes and changes in the concentration of sex hormones in various phases of the reproductive cycle may also affect the biochemical and metabolic processes of the body, which may be important in shaping the parameters that determine the quality of the obtained meat raw material.

The aim of the study was to assess the effect of the level of selected sex hormones present in the blood serum of gilts on the haematological properties of blood, slaughter parameters, and the quality of gilt meat.

## 2. Materials and Methods

The study was carried out in one of the slaughterhouses located in south-eastern Poland on a population of 60 gilts slaughtered. The processed gilts which were intended to be used for research were selected in the holding pens. For the study, gilts of the Polish Landrace breed with a body weight of over 125 kg were selected. The pigs were not all from one farm and were selected on the basis of pre-slaughter weighing in the livestock warehouse. Organisms that achieved such a weight should be characterized by a high level of maturity and therefore were affected by the sexual cycle. Before slaughter, the pigs were stunned by means of electric impulse. Slaughter was conducted in the winter season. During slaughter, bleeding was performed in a hanging position, and blood samples were collected, in which haematological and biochemical indicators were determined. For this test, blood flowing from the slaughter wound was collected by trained personnel within 20 s of the onset of bleeding. Two blood samples per animal were taken. Blood for haematology tests was collected into calibrated tubes sprayed with an anti-coagulant (double-potassium EDTA—Sigma Aldrich).

During the investigation, HGB (haemoglobin), RBCs (red blood cells expressed as a cell number present in 1L of blood sample), WBCs (white blood cells expressed as a cell number present in 1L of blood sample), PCV (packed cell volume), MCV (mean corpuscular volume), HCT (haematocrit), PLT (an indicator of the amount of platelets in the blood), PCT (plateletcrit), MCH (mean corpuscular haemoglobin), MCHC (mean corpuscular haemoglobin concentration), RDW (red blood cell distribution width), MPV (Mean Platelet Volume), PDW (Platelet distribution width) LYMPH % (percentage of lymphocytes), MONO % (percentage of monocytes), NEUT % (percentage of neutrophiles), LYMPH # (total count of lymphocytes 10^9^/L), MONO # (total count of monocytes 10^9^/L), and NEUT # (total count of neutrophiles 10^9^/L) haematological indices were determined in blood. The methods used during the research have previously been adopted in clinical trials on pigs, which were conducted by Winnicka [14]. The determinations were made using the MINDRAY BC-30vet apparatus. Blood for biochemical determinations was collected into dry test tubes filled with granules for rapid clotting. In order to obtain serum for biochemical tests, blood was centrifuged for 20 min at a speed of 3000 rpm at a temperature of 4 °C. The obtained plasma, after dispensing into plastic test tubes with a capacity of 3.0 mL, was stored at the temperature of −25 °C. In the blood serum, the content of total protein, glucose, levels of progesterone (P4), and luteinizing hormone (LH) were determined. The concentration of progesterone (P4) and luteinizing hormone (LH) was determined with the ECLIA electrochemiluminescence technique using the COBAS e-411 analyser, in accordance with the producers′ instruction manual. Haematological and biochemical blood tests were carried out in the Lawet Analytical Laboratory, which performs animal blood tests.

After the slaughtering operations were completed at the classification stand, the carcasses were weighed with an accuracy of 0.5 kg and the meat content was determined using the Ultra Fom 300 apparatus (SFK Technology A/S, Herlev, Denmark). The device estimates the meat content based on the thickness of the backfat and the longissimus dorsal muscle between the 3rd and 4th rib (counting from the caudal side at a distance of 6–7 cm from the line of carcass cut into half carcasses). Then, the initial pH_1_ of the meat was measured 45 min after bleeding. Measurements were taken in the loin (*musclus longissimus thoracis*) and ham (*semimembranosus muscle*). pH was measured at a depth of 2.5 cm with a CPU-Star pH meter (Matthäus, Pöttmes, Germany) equipped with a dagger-combined measuring cell. At the same time, the temperature T1 was measured in both muscles. The measurement was made with an electronic thermometer ET-200 with a dagger sensor. Ninety minutes after bleeding, the electrical conductivity of PE_90_ was determined using the LF STAR (Matthäus, Pöttmes, Germany) conductometer. Final pH_24_ was measured 24 h after slaughter. Then, measurements of the backfat thickness were carried out at five points in accordance with the SKURTCH methodology described in Polish, which is similar to methods utilized in the pig slaughter performance testing station [15].

Subsequently, muscle samples were taken to determine the degree of post-mortem bleeding and the content of intramuscular fat. The post-mortem bleeding of carcasses was assessed for the samples of the neck muscles (*musculi colli*) and diaphragm muscle (*musculus diaphragma*) taken from the left half of the carcass. The degree of muscle bleeding was determined by the haemoglobin-agar diffusion test as described by Beutling [16]. The test results were expressed on a 3-point scale: 1 point for a lack of a haemoglobin ring (full bleeding), 2 points for the presence of a haemoglobin ring up to 2 mm wide (incomplete bleeding), and 3 points for the presence of a haemoglobin ring more than 2 mm wide (no bleeding). The content of intramuscular fat was determined in the longissimus muscle using the Soxhlet method according to PN-ISO 1444: 2000, in the VELP Extractor 148/3 apparatus.

Statistical analysis was performed to determine how the concentration of progesterone (P4)—low level (less than 10 ng/mL), high level (greater than 10 ng/mL)—and luteinizing hormone (LH)—low level (less than 5 mlU/mL), high level (above 5 mlU/mL)—affect the haematological and biochemical blood parameters as well as the parameters of the slaughter value and meat quality. Due to the incompatibility of the distribution of the analysed variables with the normal distribution and due to the comparison of non-parallel groups, the non-parametric Mann–Whitney U test was used. The level of differences between the groups of observations was calculated using the coefficient r = $\frac{z}{\sqrt{n}}$. Statistical calculations were made using Statistica version 12.

## 3. Results

Figure 1 and Figure 2 show the results of individual measurements of progesterone and LH concentrations in the blood serum of the examined gilts. Analysis of the data shows that the concentration of progesterone ranged from 0.20 ng/mL to 40.82 ng/mL. According to the adopted scheme, the group of gilts with high P4 concentration included individuals whose blood level exceeded 10 ng/mL. Of the 60 gilts whose blood was taken for testing, high levels of progesterone were recorded in 18 samples.

On the other hand, high (above 5 mlU/mL) concentrations of LH hormone were found below the limit of quantification related to the sensitivity of the measurement equipment in the serum collected from 10 animals.

The results of the statistical evaluation of haematological and biochemical blood parameters of the examined gilts are presented in Table 1.

The conducted analysis showed that in the tested blood samples, the concentration of progesterone had a statistically significant impact only on the level of total protein, which was higher in the blood samples of gilts with a low concentration of P4 (progesterone). It is also worth noting that in the blood of gilts characterized by low levels of progesterone, a slightly higher mass of haemoglobin in the red blood cell, MCH, and the number of leukocytes was observed. On the other hand, the other analysed blood parameters had higher values in the group of gilts with a higher concentration of progesterone. Nevertheless, all of the analysed parameters were within the ranges of reference values for pig blood (Table 1).

Statistical analysis using the comparative U Mann–Whitney tests revealed that the average carcass weight for both compared groups separated on the basis of progesterone concentration was similar and amounted to around 100 kg (Table 2). Nevertheless, it is worth noting the high value of the standard deviation of this parameter, which in the group with a low level of P4 was 15.44 kg, and in the group with a high concentration of P4, it was 14.40 kg (Table 2). The analysis of the data indicates that the concentration of progesterone had a statistically significant impact on the carcass meatiness index (*p* < 0.05; r = 0.56). A lower percentage of meat content index in the carcass, on average 54.13%, was found in gilts with a high concentration of progesterone. It is also worth noting that gilts’ carcasses from this group were also characterized by a lower average backfat thickness determined on the basis of five measurements. The difference in backfat thickness between the compared groups was on average 1.16 mm.

The conducted research showed that the concentration of progesterone did not have a statistically significant effect on the physicochemical parameters of the meat of the examined gilts. Both loin and ham muscles of gilts with a higher concentration of the analysed hormone were characterized by a higher post-slaughter temperature, higher pH_45_ values, and higher electrical conductivity of the meat. The evaluation of the degree of bleeding of the examined carcasses showed that the muscles of the gilts with a higher concentration of P4 in the blood were better bled (see Table 3).

The comparative analysis performed to examine the effect of LH concentration on the variables specified in the tests showed that the LH hormone concentration had a statistically proven effect on a number of variables in both the group of blood biochemical and haematological indicators (Table 4) and the post-slaughter meat quality indicators (Table 5 and Table 6).

Detailed analysis of the obtained results revealed that the concentration of LH hormone had a statistically confirmed effect in the case of the tested samples, e.g., on the levels of WBC Z = 2.75, *p* < 0.01, r = 0.43; RBC Z = 2.18, *p* < 0.05, r = 0.35; HGB Z = 2.18, *p* < 0.05, r = 0.35; HCT Z = 4.06, *p* < 0.001, r = 0.64; MCVZ = 4.37, *p* < 0.001, r = 0.69; MCHC Z = 4.46, *p* < 0.001, r = 0.71; MPV Z = 3.45, *p* < 0.01, r = 0.57; PDW Z = 4.22, *p* < 0.001, r = 0.70; MONO Z = 3.99, *p* < 0.001, r = 0.66; MONO Z = 3.29, *p* < 0.01, r = 0.54; and NEUT Z = 2.72, *p* < 0.01, r = 0.45. In the case of samples with LH hormone concentrations above 5 mlU/mL, significantly lower mean WBC, RBC, HGB, HCT, MCV, PDW, and NEUT values and significantly higher mean MCHC, MPV, and MONO concentrations were observed. Based on the value of the coefficient r, it can be concluded that the concentration of lutropin had the greatest impact on the level of MCHC, MCV, and PDW (Table 4).

The results concerning the impact of the LH hormone level on the analysed meat quality parameters of the assessed gilts seem particularly interesting. It was found that carcasses of gilts with higher levels of the hormone were characterized by a meat content index that was lower by nearly 3% (see Table 5). It was shown that the concentration of LH affected the post-slaughter temperature of the sirloin and ham muscles: Z = 2.97, *p* < 0.01, r = 0.38, and Z = 4.20, *p* < 0.001, r = 0.54, respectively. In both cases, it was found that the muscles of gilts with higher LH concentrations in the blood were characterized by a higher temperature In further analysis of the obtained data, the significant effect of the analysed factor on the pH_45_ of the ham (Z = 2.02, *p* < 0.05, r = 0.26), which was significantly lower in gilt carcasses with lower LH levels, was worth noting. However, the pH_45_ value of the tenderloin in both groups was almost identical (Table 6).

Particularly interesting are the results of carcass bleeding tests, which showed a strong relationship between the level of bleeding and the concentration of the analysed hormone: Z = 3.27, *p* < 0.01 neck muscles, r = 0.42, and Z = 3.45, *p* < 0.01, r = 0.45, diaphragm (Table 6). Carcasses of gilts with low blood LH levels were definitely better bled. In both cases, a statistically highly significant influence of the experimental factor on the carcass bleeding was confirmed (Table 6). However, the concentration of LH hormones had the greatest effect on ham temperature (r = 0.54).

## 4. Discussion

The analysis of the literature data shows that the level of some parameters of slaughter carcasses and quality parameters of pork may be related to the body′s hormonal balance. In the study [17], particular attention was paid to the assessment of the impact of stress hormones, cortisol, and catecholamines (adrenaline and noradrenaline) on the qualitative characteristics of pig carcasses. Foury et al. [4] obtained negative correlations between the concentration of cortisol and the content of meat in the carcass, as well as positive correlations between the concentration of catecholamines and the pH 24 of meat. These studies also showed a link between the levels of stress hormones (cortisol and catecholamines) in the urine and carcass composition and muscle quality, which, according to the authors, can be explained by the effects of these hormones on energy and protein metabolism. In turn, [18] found that the administration of growth hormone to porkers accelerates the growth rate, increases the weight of valuable carcass cuts, and reduces the level of subcutaneous fat without significantly affecting the content of intramuscular fat. From the group of sex hormones, skatole and androsterone are of particular importance for shaping the quality characteristics of pork. Their high levels in male meat are responsible for unfavourable changes in the smell of meat and fat which are not accepted by consumers [3,9,19]

In the literature reports on the subject of research, there are no data on the influence of changes in the hormonal balance that occur in females during the reproductive cycle on the quantitative and qualitative characteristics of pig carcasses. In the pork processing industry, especially in the group of processors specializing in the craft production of traditional pork products, there is a well-established view that due to the deterioration of processing qualities, gilts showing symptoms of oestrus should not be sent to slaughter. In the research presented in this paper, an attempt was made to verify the hypothesis that the phases of the reproductive cycle with characteristic changes in the concentration of sex hormones affect the quality features of carcasses. In particular, dynamic changes in the production and level of sex hormones were observed during oestrus and ovulation. Oestrus in female pigs lasts on average 40–60 h. As reported by [10] the duration of heat varies over the range from 24 to 90 h, which is caused by individual variability between organisms and the level of stress. The observation of a change in LH hormone concentration is considered a sign of oestrus and ovulation symptoms [12]. The studies by [20] show that pre-ovulatory LH release coincides with the appearance of the first oestrus symptoms. The maximum concentration of LH occurs in the period of 8-24 h from the onset of the oestrus.

The nature of the hormonal changes that occur in the gilts allows us to conclude that the gilts examined in our research were slaughtered at various stages of the reproductive cycle. In particular, it should be emphasized that some of the animals were slaughtered in the oestrus phase. This is indicated by the high values of the LH hormone, which were recorded in ten individuals (Figure 2). LH hormone release is observed only in the oestrus phase [12].

In this study, the basic biochemical and haematological parameters of the blood of slaughtered gilts were determined simultaneously with the concentration of progesterone and luteinizing hormone. The results of these studies are presented in Table 1 and Table 4. The analysis of the biochemical parameters of the blood serum in pigs is used to determine the level of metabolic changes and the state of the organism [21]. According to [22], haematological and biochemical parameters of blood are of great importance for the assessment of health status in the diagnosis of diseases and the growth and nutritional status of pigs. For clinicians and researchers, it is crucial to determine and constantly update reference intervals (RIs) for individual blood parameters, the range of which is based on a specific percentage (usually 95%) of the healthy animal population. At the same time, according to the researchers of [23], the high variability of biological features occurring in pigs used in different geographical areas, as well as within individual subpopulations, is the factor limiting the determination of universal reference intervals for haematological and biochemical parameters.

This study demonstrates that most of analysed blood parameters of the tested animals were within the reference values for the population of pigs bred and slaughtered in Poland [14]. It should be emphasized that all the gilts selected for the study did not show any visible external clinical signs or physical abnormalities, did not come from ASF-infected areas and, on the basis of a health certificate issued by a veterinarian, could be sent for slaughter in accordance with the applicable sanitary procedures. Despite the fact that the tests did not show that the norms of the determined blood parameters were exceeded, it should be noted that there were significant intergroup differences in blood parameters related to the level of the LH hormone (Table 4). Interpretation of these results is difficult due to the lack of data in the literature. It can be assumed that the demonstrated differences could have a physiological basis related to the phases of the reproductive cycle [24]. According to Friendship and Henry [21], depending on the age of the females and the phase of the reproductive cycle, some haematological indicators change their activity, but there are also those whose concentration remains relatively constant. The fluctuations noted in our own research could also be the effects of stress, animal excitement, the handling process, and disturbances in the homeostasis of the organism in the pre-slaughter period [23,25].

The results obtained in this study indicate that the examined gilts were characterized by a high and diversified carcass weight. This is indicated by the high value of the arithmetic mean and the value of the standard deviation of the carcass weight in all compared groups (Table 2 and Table 5). It is worth noting that, in many countries, there has been a significant trend of increasing the body weight of slaughtered pigs in recent years. According to Shahbandeh [26], in Great Britain in the years 2003–2021, the average carcass weight increased by 12.8 kg from 74.1 kg in 2003 to 88.6 kg in 2021. A long-term trend of increasing the weight of market porkers and carcasses is also noted in the USA.

According to data from the Economic Research Service U.S. Department of Agriculture, in 1989–2019, the weight of pig carcasses increased by as much as 19% from 81.2 to 96.8 kg [27]. The increase in body weight observed on a global scale indicates that older, and thus more physiologically and sexually mature, animals are selected for slaughter. The increase in the weight of slaughtered pigs and the associated weight index of the produced pig carcasses have significant technological and economic consequences [28]. The review [29] shows that the increase in the weight of fattening pigs causes significant changes in the composition of the carcass, its linear parameters, and the quality parameters of the meat.

The authors showed, inter alia, that an increase in weight by 10 kg results in an average increase in backfat thickness by 1.8 mm, with a simultaneous decrease in the efficiency of valuable meat cuts. Attention was also paid to changes in quality parameters, including colour brightness, initial and final pH, and free leakage with increasing weight of porkers. According to Kapelański and Grajewska [6], in the meat of porkers slaughtered at a higher weight, and therefore older and more mature, there are fewer quality defects of the PSE type. According to Van den Broeke et al. [28], not only is the weight of the carcass important, but so is value of the kilogram of the carcass, which depends on the processing suitability. The reduction in the value of heavy carcasses is influenced by the reduced meat content and higher fat content. At the same time, heavier carcasses have better processing potential [6]. However, due to economic criteria taking into account production costs, fatteners for slaughter should be directed before reaching the weight of 130 kg [28].

The analysis of the basic slaughter indicators shows that gilts with low levels of progesterone and LH in the blood were characterized by a significantly higher meatiness index and a lower mean fat thickness (Table 5). Obviously, the demonstrated difference could not be the result of changes in the concentration of hormones related to the phases of the reproductive cycle. The meatiness of the carcasses, like the fatness, is a parameter determined by the long-term influence of a wide range of factors related to the genetic potential of animals and the fattening system. At the same time, the literature [30,31,32,33] provides data on the significant influence of male sex and stress hormones on the quantitative and qualitative parameters of pig carcasses.

In this study, significant differences were noted in most of the analysed quality parameters of meat related to the level of LH hormone concentration. First of all, it should be emphasized that there were differences in muscle temperature, which was higher by 2 °C in the group of gilts with high LH levels (Table 6). Previous studies [34] show that in the pre-ovulation period in pigs, an increase in body temperature is observed, the maximum values of which are recorded within 24–48 h from the appearance of the tolerance reflex in gilts. According to the authors, the increase in temperature is associated with higher oestradiol secretion, the peak of which occurs 24 to 48 h before oestrus and precedes the pre-ovulatory LH release.

As can be assumed, higher body temperature, which is one of the symptoms of heat, may be related to the higher temperature of muscles in the gilts, in whose blood the LH concentration was higher, as reported in our research. High muscle temperature was probably the result of hormonal interactions and the result of thermal stress, which is often observed in animals during the pre-slaughter period [25]. Increased muscle temperature in animals with higher LH levels could also be associated with high physical activity in gilts showing oestrus symptoms. According to Johnson et al. [13], one of the behavioural symptoms accompanying oestrus is increased physical activity, including the state of muscle tension and the jumping reflex of other animals. Oestrus behaviour causes physical fatigue in the body. This could result in a reduction in energy resources and disturbances in the course of post-mortem glycogenolysis [25]. In this study, in the group with a higher LH level, a decrease in the degree of post-slaughter muscle bleeding was also noted (Table 6), which is one of the symptoms of physical fatigue in slaughter animals [35].

## 5. Conclusions

The aim of the conducted research was to determine the impact of factors such as sex hormone levels, which vary during gilts’ sexual cycles, on the quality of the obtained meat and slaughter characteristics of the processed gilts. In general, in cases of increased levels of the LH, it was revealed that there is a dependence between this sex hormone level and higher body temperature. The meat quality and slaughter characteristics were also affected. The significant effect of the analysed factor on the pH_45_ of the ham, which was significantly lower in gilt carcasses with lower LH levels, was worth noting. It was found that carcasses of gilts with higher levels of the hormone were characterized by a meat content index that was lower by nearly 3%. It was shown that the concentration of LH affected the post-slaughter temperature of the sirloin and ham muscles. Particularly interesting are the results of carcass bleeding tests, which showed a strong relationship between the level of bleeding and the concentration of the analysed hormone. Carcasses of gilts with low blood LH levels were definitely better bled.

## Figures and Tables

**Figure 1 animals-13-00267-f001:**
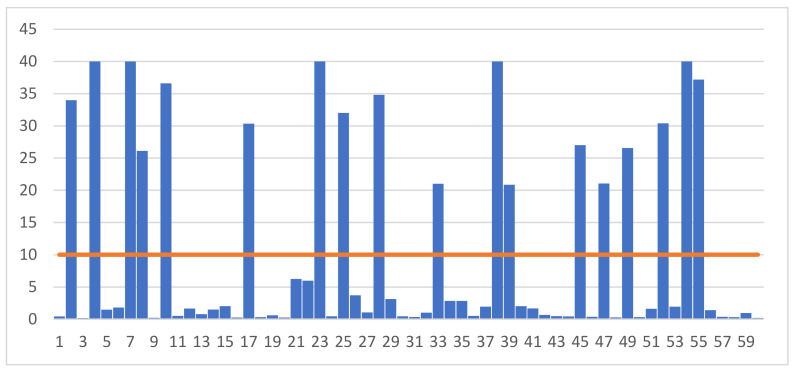
Individual results of measuring the concentration of progesterone (P4) (ng/mL) in the blood of the examined gilts.

**Figure 2 animals-13-00267-f002:**
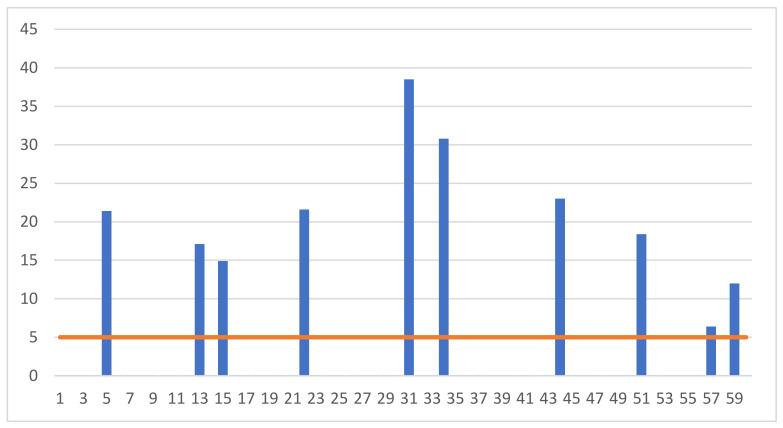
Individual results of measuring the concentration of luteinizing hormone (LH) (mlU/mL) in the blood of the examined gilts.

**Table 1 animals-13-00267-t001:** Influence of progesterone level (ng/mL) on selected haematological and biochemical blood parameters of gilts.

	Progesterone < 10		Progesterone ≥ 10		Z	*p*	r
	M	SD	M	SD			
WBC (10^9^/L)	15.53	3.86	17.79	2.95	1.62	0.105	0.21
RBC (10^12^/L)	7.86	0.72	8.05	0.44	0.64	0.520	0.10
HGB (g/dL)	13.84	1.12	13.94	1.69	0.49	0.624	0.08
HCT (%)	44.69	5.67	44.01	6.45	0.41	0.685	0.06
PLT (10^9^/L)	323.97	98.77	378.50	145.67	0.96	0.335	0.15
PCT (%)	0.39	0.13	0.53	0.26	0.92	0.359	0.14
MCV (µm^3^ fL)	56.84	4.27	54.50	5.50	1.09	0.278	0.17
MCH (pg)	17.65	0.89	17.28	1.57	0.44	0.660	0.07
MCHC (g/dL)	31.18	2.42	31.84	2.28	0.58	0.565	0.09
RDW (%)	17.33	1.27	17.58	1.04	0.78	0.436	0.12
MPV (fL)	10.71	1.65	11.74	2.08	1.13	0.261	0.19
PDW (fL)	8.51	4.64	7.64	4.70	0.51	0.607	0.09
LYMPH (%)	40.42	9.65	39.74	12.11	0.78	0.437	0.12
MONO (%)	10.23	4.54	8.48	5.09	0.83	0.406	0.14
NEUT (%)	48.64	11.18	51.79	12.30	0.81	0.417	0.13
LYMPH (10^9^/L)	5.88	2.08	7.15	3.16	1.03	0.302	0.16
MONO (10^9^/L)	1.50	0.61	1.38	0.77	0.59	0.554	0.10
NEUT (10^9^/L)	7.79	3.21	9.26	2.76	1.57	0.117	0.26
Glucose (mg/dL)	105.41	17.70	110.67	42.74	0.42	0.672	0.09
P-LCR (%)	26.72	4.05	24.50	4.69	0.74	0.459	0.19
Total protein (g/dL)	6.67	0.33	5.73	0.67	2.50	0.012	0.56

M—mean; SD—standard deviation; Z—Mann–Whitney U statistics; *p*—level of statistical significance. The level of differences between the groups of observations was calculated using the coefficient r = $\frac{z}{\sqrt{n}}$. Statistical calculations were made using Statistica version 12.

**Table 2 animals-13-00267-t002:** Influence of progesterone level (ng/mL) on selected slaughter parameters of gilts carcasses.

	Progesterone < 10		Progesterone ≥ 10		Z	*p*	r
	M	SD	M	SD			
Body weight (kg) Carcass weight (kg)	128.88 100.66	16.01 15.44	127.82 99.97	13.99 14.40	0.26 0.05	0.74 0.957	0.32 0.01
Meatiness (%)	56.82	2.71	54.13	4.24	2.32	0.021	0.30
Backfat thickness (mm)	22.98	3.98	24.14	4.56	0.69	0.490	0.09
Intramuscular fat content (%)	2.64	1.75	2.48	2.20	0.28	0.777	0.08

M—mean; SD—standard deviation; Z—Mann–Whitney U statistics; *p*—level of statistical significance. The level of differences between the groups of observations was calculated using the coefficient r = $\frac{z}{\sqrt{n}}$. Statistical calculations were made using Statistica version 12.

**Table 3 animals-13-00267-t003:** Influence of progesterone level (ng/mL) on selected quality parameters of gilts meat.

	Progesterone < 10		Progesterone≥ 10		Z	*p*	r
	M	SD	M	SD			
Temp._45_ loin (°C)	37.49	1.87	36.92	1.66	1.31	0.190	0.17
Temp._45_ ham (°C)	39.28	1.39	38.59	2.05	0.78	0.435	0.10
pH_45_ loin (pH)	6.20	0.24	6.04	0.41	1.72	0.085	0.22
pH_45_ ham (pH)	6.17	0.28	6.16	0.31	0.18	0.858	0.02
PE_90_ loin (mS)	3.68	0.75	3.53	0.59	0.88	0.379	0.11
PE_90_ ham (mS)	3.92	1.08	3.81	0.54	0.06	0.956	0.01
pH_24_ loin (pH)	5.68	0.32	5.54	0.23	1.60	0.110	0.21
pH_24_ ham (pH)	5.68	0.28	5.65	0.31	1.12	0.262	0.14
PE_90_ loin (mS)	4.44	1.58	4.31	1.68	0.48	0.634	0.06
PE_90_ ham (mS)	4.48	1.78	5.15	1.75	1.30	0.193	0.17
Evaluation of bleeding of the colli muscle (point)	1.44	0.54	1.19	0.48	1.57	0.116	0.20
Evaluation of bleeding of the diaphragm muscle (point)	1.49	0.73	1.15	0.38	1.47	0.143	0.19

M—mean; SD—standard deviation; *Z*—Mann–Whitney U statistics; *p*—level of statistical significance. The level of differences between the groups of observations was calculated using the coefficient r = $\frac{z}{\sqrt{n}}$. Statistical calculations were made using Statistica version 12.

**Table 4 animals-13-00267-t004:** Effect of luteinizing hormone level (mlU/mL) on selected haematological and biochemical blood parameters of gilts.

	LH < 5		LH ≥ 5		Z	*p*	r
	M	SD	M	SD			
WBC (10^9^/L)	17.94	3.89	14.68	3.13	2.75	0.006	0.43
RBC (10^12^/L)	8.14	0.70	7.73	0.61	2.18	0.029	0.35
HGB (g/dL)	14.29	1.30	13.57	1.12	2.18	0.029	0.35
HCT (%)	48.91	4.59	41.65	4.51	4.06	0.000	0.64
PLT (10^9^/L)	320.00	126.01	344.79	99.11	0.80	0.423	0.13
MCV (fL)	60.13	2.16	53.88	4.00	4.37	0.000	0.69
MCH (pg)	17.55	0.66	17.59	1.26	0.35	0.730	0.05
MCHC (g/dL)	29.23	0.80	32.70	2.03	4.46	0.000	0.71
RDW (%)	17.10	1.47	17.56	1.01	1.87	0.062	0.30
MPV (fL)	9.86	0.46	11.54	1.95	3.45	0.001	0.57
PDW (fL)	12.64	0.88	5.87	3.99	4.22	0.000	0.70
LYMPH (%)	39.33	11.42	40.92	9.17	0.64	0.525	0.10
MONO (%)	5.78	2.44	12.05	4.04	3.99	0.000	0.66
NEUT (%)	53.55	12.86	47.03	9.96	1.53	0.127	0.25
LYMPH (10^9^/L)	7.02	2.65	5.55	1.95	1.75	0.079	0.28
MONO (10^9^/L)	1.03	0.39	1.71	0.62	3.29	0.001	0.54
NEUT (10^9^/L)	10.08	3.56	7.04	2.33	2.72	0.007	0.45
Glucose (mg/dL)	106.06	24.04	106.75	4.65	0.62	0.539	0.14
P-LCR (%)	26.46	4.18	25.63	4.57	0.34	0.737	0.08
Total protein (g/dL)	6.46	0.53	6.83	0.28	1.47	0.141	0.33

M—mean; SD—standard deviation; Z—Mann–Whitney U statistics; *p*—level of statistical significance. The level of differences between the groups of observations was calculated using the coefficient r = $\frac{z}{\sqrt{n}}$. Statistical calculations were made using Statistica version 12.

**Table 5 animals-13-00267-t005:** Influence of luteinizing hormone level (mlU/mL) on selected slaughter parameters of gilt carcasses.

	LH < 5		LH ≥ 5		Z	*p*	r
	M	SD	M	SD			
Body weight Carcass weight (kg)	130.12 102.66	18.41 17.35	126.9 97.88	12.43 11.58	2.41 1.41	0.162 0.158	0.31 0.18
Meatiness (%)	57.30	3.13	54.94	2.97	3.47	0.001	0.45
Backfat thickness (mm)	22.51	4.73	24.12	3.02	1.57	0.117	0.20
Intramuscular fat content (%)	2.76	1.91	2.00	1.47	0.70	0.484	0.19

M—mean; SD—standard deviation; Z—Mann–Whitney U statistics; *p*—level of statistical significance. The level of differences between the groups of observations was calculated using the coefficient r = $\frac{z}{\sqrt{n}}$. Statistical calculations were made using Statistica version 12.

**Table 6 animals-13-00267-t006:** Effect of luteinizing hormone level (mlU/mL) on selected quality parameters of gilt meat.

	LH < 5		LH ≥ 5		Z	*p*	r
	M	SD	M	SD			
Temp. loin (˚C)	36.04	2.18	38.04	1.14	2.97	0.003	0.38
Temp. ham (˚C)	38.10	1.63	39.07	0.75	4.20	0.000	0.54
pH_45_ loin (pH)	6.17	0.29	6.16	0.29	0.04	0.970	0.00
pH_45_ ham (pH)	6.09	0.29	6.26	0.24	2.02	0.043	0.26
PE_90_ loin (mS)	3.79	0.79	3.47	0.56	1.46	0.144	0.19
PE_90_ ham (mS)	4.17	1.05	3.54	0.77	3.10	0.002	0.40
pH_24_ loin	5.53	0.12	5.80	0.39	3.67	0.000	0.47
pH_24_ ham	5.56	0.11	5.81	0.36	3.15	0.002	0.41
PE_90_ loin (mS)	4.75	1.87	4.00	1.05	0.90	0.368	0.12
PE_90_ ham (mS)	4.60	1.99	4.66	1.52	0.42	0.672	0.05
Evaluation of bleeding of the colli muscle	1.18	0.39	1.63	0.58	3.27	0.001	0.42
Evaluation of bleeding of the diaphragm muscle	1.13	0.34	1.76	0.82	3.45	0.001	0.45

M—mean; SD—standard deviation; Z—Mann–Whitney U statistics; *p*—level of statistical significance. The level of differences between the groups of observations was calculated using the coefficient r = $\frac{z}{\sqrt{n}}$. Statistical calculations were made using Statistica version 12.

## Data Availability

Not applicable.

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
