# Peer review of "Influence of the Level of Sex Hormones in the Blood of Gilts on Slaughter Characteristics and Meat Quality"

_animals, 2023, doi:10.3390/ani13020267_

Round 1
Reviewer 1 Report
The paper would also be interesting, however it requires major changes to be acceptable for publication:
- In the article, including the title, the authors use the term gilts (for Directive 2008/120 it means "pig female after puberty and before farrowing"); in line 44 the term "breeding gilts" is used. however I assume that in Poland, in the pork processing industry (as written in lines 294-295) "fattening female" and not "breeding gilts" is used
- the introduction needs to be completely revised. it is clear that this is a consequence of copy-pasting with repeated statements (see lines 30-38 vs 44-50 and lines 39-43 vs 51-59).
- If for progesterone the experimental design is considered adequate (37 animals vs 23 animals), as regards the lutropin concentrations this is not acceptable (10 vs 50) and I believe that the results are strongly influenced by the small number of samples with concentrations > 5mlU/ml. In my opinion these results should be deleted from the document.
-The discussion must be reduced. The whole part of reproductive physiology (lines 275-320) can be eliminated
- in the bibliography there are numerous quotations in Polish, which cannot be verified. perhaps better explain the methodologies in the text.
The paper may be re-evaluated after being suitably reformulated
Author Response
Dear reviewer below the authors prepared reactions and answers to your valuable comments on the manuscript. The authors are grateful for the resulting improvement of our work.
The answers for your comments were placed below the relevant text fragments and were distinguished with red font.

Reviewer 2 Report
The authors aimed to investigate the possible impact of P4 and LH levels, which vary during gilts reproductive cycle, on the slaughter characteristics and meat quality.
The manuscript is attractive for readers not only in the field but also for industry groups (especially the pork processing industry) and consumers; it's interesting especially because there isn't much literature data on the explanation of the differences in the quality parameters of carcasses observed within individual sexes. Particularly interesting are the results of LH affected the post-slaughter temperature of the sirloin and ham muscles.
However, a few typographical errors can be found in the text:
Line 74: Please clarify this sentence. Do you mean “Changes in the level of sex hormones come with characteristic symptoms…”?
Lines 91-92: Please rephrase the sentence“The research material that was used during tests was a population of 60 gilts slaughtered.” as follow “The study was carried out in one of the slaughterhouses located in south-eastern Poland on a population of 60 gilts slaughtered”.
Lines 93-94; 95-96: Authors should describe the stunning method used. Blood samples were collected by properly trained personnel? If the answer is yes, please indicate it. The number of samples collected per group is unclear. Please indicate this information.
Lines 132-133: The sentence may be omitted because the concept has just been expressed.
Line 155: Until this point, authors used “luteinizing hormone (LH)” referring to LH, in this line “lutropin”. Authors should choose a unique form indicating the LH hormone and after the first mention of the extended form they should use the short in the rrest of the text.
Line 204: please add “s” to “gilt”.
Line 249-250: Please rephrase the sentence in the following way “In both cases, it was found that the muscles of gilts with higher LH concentrations in the blood were characterized by a higher temperature.”
Lines 250-253: Please add an “s” to “gilt”; Authors should add an explicative quotation into the text and the references referring to “In further analysis…” .
Line 263: please add “:” after “hormone”.
Line 284: Authors should add the quotation they referred to in the text.
Line 383: “(performance 1, performance 2).” Can you explain what are you referring to with these brackets?
Lines 420-423: Authors shoul delete these two sentences not to be repetitive.
Line 427 : Please choose and check the character style of pH45 in the text and in Table 6 too.
The tables are appropriate and easy to interpret.
The cited references are mostly recent publications and relevant. The following references could be included in the Discussion [for example in (lines 275-291) and (lines 354-357)] for completeness of content:
- Bozzo G, Padalino B, Bonerba E, Barrasso R, Tufarelli V, Zappaterra M, Ceci E. Pilot Study of the Relationship between Deck Level and Journey Duration on Plasma Cortisol, Epinephrine and Norepinephrine Levels in Italian Heavy Pigs. Animals (Basel). 2020 Sep 4;10(9):1578. doi: 10.3390/ani10091578
- Ceci E, Marchetti P, Samoilis G, Sportelli S, Roma R, Barrasso R, Tantillo G, Bozzo G. Determination of plasmatic cortisol for evaluation of animal welfare during slaughter. Ital J Food Saf. 2017 Sep 29;6(3):6912. doi: 10.4081/ijfs.2017.6912.
- Bozzo G, Barrasso R, Marchetti P, Roma R, Samoilis G, Tantillo G, Ceci E. Analysis of Stress Indicators for Evaluation of Animal Welfare and Meat Quality in Traditional and Jewish Slaughtering. Animals (Basel). 2018 Mar 21;8(4):43. doi: 10.3390/ani8040043
English language is appropriate enough and understandable; the conclusions are consistent with the evidence and arguments presented.
Taking all this into account the work fits the journal aims.
Author Response

(The authors gave the same response as above.)

Reviewer 3 Report
REVIEW OF MANUSCRIPT: Manuscript ID: animals-2100340 v0
Type of manuscript: Article
Title:
Influence of the level of sex hormones in the blood of gilts on slaughter characteristics and meat quality
|
Criteria for evaluation |
|
|
|
|
|
Ratings: |
EXCEL/HIGH |
GOOD |
FAIR |
POOR |
|
Originality/Novelty: |
X |
|
|
|
|
Significance: |
|
X |
|
|
|
Quality of presentation: |
|
X |
|
|
|
Scientific Soundness: |
|
X |
|
|
|
Importance /Interest to the Readers: |
|
X |
|
|
|
Overall Merit: |
|
X |
|
|
|
Language quality: |
|
X |
|
|
|
Citations: |
|
X |
|
|
|
International relevance: |
|
|
X |
|
|
|
|
|
|
|
The aim of the conducted research was to determine the impact of factors such as sex hormones levels which vary during gilts reproductive cycle on the quality of the obtained meat and slaughter characteristics of the processed gilts. The research material included a population of 60 gilts slaughtered in one of the slaughterhouses located in south-eastern Poland. After the slaughtering operations were completed, the carcasses were weighed at the classification stand. The results of the statistical evaluation of hematological and biochemical blood parameters of the examined gilts showed that in the tested blood samples the concentration of progesterone had a statistically significant impact only on the level of total protein, which was higher in the blood samples of gilts with a low concentration of progesterone. It was found that carcasses of gilts with higher levels of the LH hormone were characterized by a meat content index lower by nearly 3%. It was shown that the concentration of LH affected the post-slaughter temperature of the sirloin and ham muscles. The interpretation of the obtained data was difficult since there seems to be a gap in the literature concerning the dependencies of sex hormone levels in gilts and meat quality.
General observations
This study has been undertaken to show the influence of sex hormones on meat quality. The article gives information on the effect of progesterone and LH on meat quality. The study is interesting from an academic point of view but has little or no value for the meat or pig industries as these are factors difficult to control on a large scale. The study is lacking some preliminary data that are important for the study as it is well known that many different factors can influence the sex hormonal levels, such as breed of gilts, age at slaughter (average age), season/month of slaughter since climate/temperature has an effect on hormone production, farm management practice (stress & feed), grouping at slaughter house and holding pens, animal welfare at slaughter, are some of the conditions to consider.
Below are the comments of the reviewer.
COMMENTS OF REVIEWER:
Originality/Novelty: The topic addressed here is important from an academic point of view and relevant to the current trend of food quality. No other study was found in the quick search that shows the links of progesterone and luteinizing hormone to meat quality.
Significance: The results have been interpreted correctly according to data available and points out to the effect of progesterone on total protein level and LH on meat content index. But under normal pig production or slaughter these values are difficult to control.
Quality of Presentation: The article is written in an appropriate way with the data and analyses presented in a form that the results could be discussed. The discussion is well written.
Scientific Soundness: The study lacks some relevant data that may be available from the slaughterhouse and pig breeding farm. These remarks are mentioned below in the section of Specific Remarks
Interest to the Readers: The article will attract more attention of the academics. For the producer or the slaughterhouse personnel it will be less interesting because of the difficulty in controlling individual hormonal levels and as the treatments are always made as a group.
Overall Merit: Is there an overall benefit to publishing this work? Does the work provide an advance towards the current knowledge? Yes this work does show a different perspective of the sex hormones on meat quality.
English Level: The English language is understandable but needs to be revised.
Specific Remarks
Simple Summary: Could be improved
Abstract: Needs improvement. There are some parts that can be altered to mention the observations with less words. Apart from the number of animals in the study (60), it is important to mention the breed of animal used, their origins, age at slaughter, their housing and feeds
INTRODUCTION:
Introduction is mostly well written with relevant references.
Line 35: Needs a reference – “increase in the meat-iness index of the carcasses” (REF)
Line 39: It would be preferable to specify if the male castrates are surgical castrated or immunocastrates. Since in the beginning the authors have mentioned that the castration is by ….. &……. Line 30-31
Line 45: male castrates – to specify one or both are used?
Paragraph Lines 61-65 – Reference is needed.
Paragraph Lines 65 – 67 – Needs restructuring.
Lines 67-68: what is the meaning of – breed of the maintenance system?
Line 87 – Check spelling of Haematological – appears as heamatological properties of blood.
MATERIAL AND METHODS
General comment: There is a major flaw here in this article. For any animal experiment or study there are some information’s that are crucial to provide. It is well known there is always an effect of genetic variation and environmental factors that vary in different farms and at different times of the year, which can influence the onset of puberty, the hormonal levels, and the quality of the meat products. Further, the method of slaughter also plays an important part in the process, mainly to avoid stress. These factors should be seen to.
Apart from the number of animals in the study (60), it is important to mention the breed of animal used, their origins, age at slaughter, their housing and feeds. Time of slaughter (season) – Heat and cold have their effect on the animal.
Line 92: What is “livestock warehouse” Are the authors referring to the holding pens for pigs in the slaughterhouse before slaughter?
Line 98: double-potassium EDTA – mention the laboratory of manufacture.
Line 101: PCV & not PVC
Line 123: Indicate the manufacturing company of Ultra Fom 300 apparatus. What is the methodology used here to measure the meat content?
Line 142: The anatomic name of the muscle should be used and not Longest thoracic muscle.
RESULTS
The data was not verified for lack of time, so please check this. On the whole it appeared to be in accordance.
Line 163 - 165: Check this paragraph. It is difficult to understand. – Language
DISCUSSION
General comment: Well written with a good discussion of the results. Further information of the effect of age at slaughter and the hormonal effect is noteworthy. Your results should be discussed with those of other authores with respect to influence of the physiological hormonal levels.
CONCLUSIONS
Needs a revision of English
Line 411: Livestock warehouse – Please check on this terminology as mentioned above
Author Response

(The authors gave the same response as above.)

Round 2
Reviewer 1 Report
No comments and suggestions for Authors in present form of the paper
Author Response
The authors are greatefull for reviewers attention and contribution.
Reviewer 3 Report
The manuscript has been improved and can be published after the changes indicated in the PDF of the article. Various correction are still to be done.
The selection of animals for the study is a bit ambiguous "body weight of over 125 kg" (What is average weight of the total of 60 gilts?). It appears that the animals were selected on a random basis and not from a group of animals with almost similar ages and weights. The selection could have been done forming groups of 10 by age and by weight.
Other comments are noted as "Notes in the text of the PDF attached in this report.

Author Response
Dear reviewer. The authors are grateful for Your attention and contribution. Abstract, Materials and methods and Discussion section were modyfied by rewriting of various senetnces and giving additional information. Spelling and aditing errors were removed.
